# Capacity configuration optimization of electricity heat hydrogen regional integrated energy system considering supply-demand uncertainties

**Tianhe Sun**©*, **Runqi Sun, Xiaoyi Qian, Baoshi Wang**

Key Laboratory of Energy Saving and Controlling in Power System of Liaoning Province, Shenyang Institute of Engineering, Shenyang, P R China

* sunth@sie.edu.cn

## Abstract

Rationally configuring the capacity of the electricity heat hydrogen regional integrated energy system is conducive to improving its economy and energy utilization efficiency. In view of the dual effects of the uncertainties of energy supply and demand in system configuration on power supply reliability and wind power consumption, a min-max-min two-stage robust optimization configuration model aiming at the minimum sum of system investment and operating cost is established for achieving an optimal capacity configuration of multi-vector technologies involved in it. On the basis of typical scenarios, a box-type uncertainty set independent of a probability distribution is used to describe the uncertainty of wind power and demand and the robustness constraint is formed. An innovative parameter, named the uncertainty adjustment parameter, is introduced into the box-type uncertainty set to avoid sacrificing economic benefits caused by too conservative configuration scheme. In this paper, the column and constraint generation algorithm and strong duality theory are used to decompose the original problem into the linearized master problem and subproblem, which can improve the solving speed. Finally, an integrated energy system in the north of China is taken as a case study. The results demonstrate that the proposed model effectively addresses the uncertainty problems of wind power and demand, leading to improved reliability and wind power integration performance. By changing the uncertainty adjustment parameter, the conservativeness of the configuration scheme can be flexibly adjusted. The effectiveness and applicability of the proposed model and solution algorithm are verified.

## 1. Introduction

To alleviate energy crises and environmental problems, the electricity heat hydrogen regional integrated energy system coupled with renewable energy generation has been developed rapidly. The integrated energy system coupling and coordinated utilization of electricity, heat, hydrogen and other energy sources can not only promote the consumption of renewable energy, but also significantly improve energy utilization efficiency [1]. However, in the

**Data availability statement:** All relevant data are within the manuscript and its Supporting information files.

**Funding:** The author(s) received no specific funding for this work.

**Competing interests:** The authors have declared that no competing interests exist.

electricity heat hydrogen regional integrated energy system, the uncertainty of renewable energy generation and demand will cause the problem of renewable energy consumption to become prominent and even make the system deviate from the optimal state. Therefore, how to rationally allocate the equipment capacity of the electricity heat hydrogen regional integrated energy system, so that the system can better deal with the uncertainty is an important problem facing the current electricity heat hydrogen regional integrated energy system.

Currently, scholars have established uncertainty optimization configuration models based on stochastic optimization [2], interval optimization [2] and robust optimization [3]. Stochastic optimization is a method of solving optimization problems with stochastic factors using tools such as probability statistics, stochastic processes and stochastic analysis. It includes methods such as scenario optimization and chance-constrained programming. Reference [4] proposes a two-stage optimization model for regional integrated energy system planning and operation considering dynamic hydrogen prices and uncertainty, describing the uncertainties of renewable energy output and demand using fuzzy chance-constrained programming, minimizing system construction and operation costs through reasonable equipment capacity configuration and economic operation planning. Reference [5] presents a multi-objective stochastic planning model that comprehensively considers distribution network operation costs, voltage stability and pollutant emissions, describing the uncertainties of renewable energy output and demand using fuzzy chance-constrained programming and selects the capacity of distributed energy sources. Reference [6] proposes a capacity optimization configuration model with the objective of minimizing the annual total cost, using ordered clustering algorithms and fuzzy C-means clustering algorithms to deal with the uncertainties of system supply-demand. Reference [7] proposes a integrated energy system capacity planning model with multiple energy flows including cooling, heating, electricity and gas, aiming to minimize the sum of investment and operating costs, using scenario analysis to describe the uncertainty of renewable energy. Stochastic optimization is an effective method for quantifying uncertainty, but it requires a large number of samples to obtain accurate probability distributions of random parameters, which are difficult and costly to determine.

Interval optimization is an optimization method based on interval analysis theory, which uses interval variables instead of traditional point variables to better handle uncertainty and data fluctuations. Reference [8] proposes a planning model for an integrated power and natural gas pipeline network from the perspective of system safety and reliability, using predicted intervals to describe the uncertainty of wind power and achieve safe and stable system operation. Reference [9] proposes interval optimization models with the objectives of minimizing investment and operating costs and maximizing renewable energy utilization, using interval numbers to represent uncertainty factors in the system. However, the results of interval optimization are usually presented in interval form rather than a specific point. This representation may lead to insufficient understanding of the exact location of the optimal solution, so interval optimization may not be ideal in scenarios where precise solutions are needed.

The goal of robust optimization is to find a solution that satisfies all constraints for all possible scenarios and minimizes the value of the objective function in the worst-case scenario. Reference [10] establishes a robust planning model for integrated energy system considering the uncertainties of multi-dimensional demands, analyzing the impact of multi-dimensional demand uncertainties on integrated energy capacity configuration. Reference [11] establishes a multi-objective robust planning model for the micro-integrated energy system with electric-thermal coupling, aiming to minimize investment and operating costs while minimizing system peak-valley fluctuations, considering the high randomness and intermittency of renewable energy output. The resulting scheme improves wind power utilization and system economics. Reference [12] introduces the theory of information gap decision-making to deal

with uncertainty on both supply and demand sides, establishing a robust optimization configuration model for integrated energy systems with the maximum internal rate of return as the optimization objective.

In summary, the following conclusions can be drawn:

The advantage of robust optimization lies in its ability to focus on the fluctuation boundaries of uncertain parameters without requiring a detailed understanding of their probabilistic distribution characteristics. It demonstrates strong resilience and is suitable for system environments where safety and reliability are the top priorities. However, to account for all possible disturbances, robust optimization may lead to relatively conservative decisions, potentially resulting in resource wastage and reduced benefits.

The advantage of interval optimization is that it only requires prior knowledge of the upper and lower bounds, or the midpoint and width, of the interval numbers. On one hand, obtaining these parameter values is typically easier. On the other hand, random variables can be converted into interval numbers through confidence levels or fuzzy numbers, thereby transforming stochastic optimization models directly into interval optimization models. This method can effectively handle uncertainty with known interval ranges and is applicable in cases where the uncertainty range of parameters in integrated energy systems is known but their specific probabilistic distributions are unclear. However, since interval optimization only considers uncertainties within the interval, the precision of its optimization results is often inferior to that of stochastic or robust optimization.

The advantage of stochastic optimization is its flexibility in handling highly random problems. It is well-suited for systems with uncertainties such as fluctuations in renewable energy generation and significant randomness in load forecasting. However, compared to other optimization methods, stochastic optimization requires substantial computational resources when addressing complex systems, leading to longer computation times and lower efficiency.

Existing studies have made some progress in addressing internal uncertainties in integrated energy systems but most of the current studies consider the impact of renewable energy uncertainty or demand uncertainty on the operation planning of integrated energy system unilaterally and the supply-demand uncertainty modeling in integrated energy system is not adaptable. Therefore, based on the above research, this paper makes the following contributions:

(1) In order to solve the problem of optimal allocation of the electricity heat hydrogen regional integrated energy system, based on typical scenarios, a box-type uncertainty set independent of probability distribution is proposed to describe the uncertainty of wind power and demand, forming robust constraints.

(2) An innovative two-stage robust optimization model is constructed. The first stage of the model is capacity configuration, and the second stage considers the operation in the worst scenario, and adopts min-max-min structure to optimize the configuration. In order to flexibly adjust the degree of conservatism according to the actual demand, the uncertainty adjustment parameter $\Gamma$ is introduced into the model. By adjusting the parameter $\Gamma$, the economy and feasibility can be more reasonably balanced, so as to provide higher adaptability for optimization in different scenarios.

(3) In order to further improve the solving efficiency of the model, the model is derived and transformed by combining the column and constraint generation algorithm and the strong duality theory, and the original problem is successfully decomposed into linearized main problem and subproblem, and efficient solution is achieved by alternate solving method.

(4) An integrated energy system in northern China was used to verify the reliability and applicability of the proposed model, and the influence of different uncertainty adjustment parameters on capacity allocation was analyzed, demonstrating the unique advantages of the model in dealing with complex uncertainties.

## 2. Preparation structure of system and mathematical model

### 2.1. Structure of system

The electricity heat hydrogen regional integrated energy system is connected to the external power grid through a common node. During normal operation, there is no power transmission on connection lines with the external power grid. Purchase electricity from the external power grid only when electric energy is insufficient to meet the demand. The system includes wind turbines, combined heat and power (CHP) units, ground-source heat pumps, power-to-hydrogen (P2H) devices and energy storage devices, *et al*. During heating season, the ground-source heat pump transfers thermal energy from soil to heating system of buildings with minimal electricity consumption, providing heating to users. Energy storage devices are used to address uncertainty of wind power. They store excess electricity generated by wind turbines and release it when wind power output is low, ensuring the system's stable operation. The P2H device converts surplus electricity into hydrogen for sale directly when there is an excess of wind power. The system couples the electricity network and heating network through CHP units. The system structure is illustrated in Fig 1.

The back-pressure CHP unit can fully utilize the steam generated by the boiler and its energy utilization efficiency can reach 80% to 85%. This type of unit performs well in energy recovery,

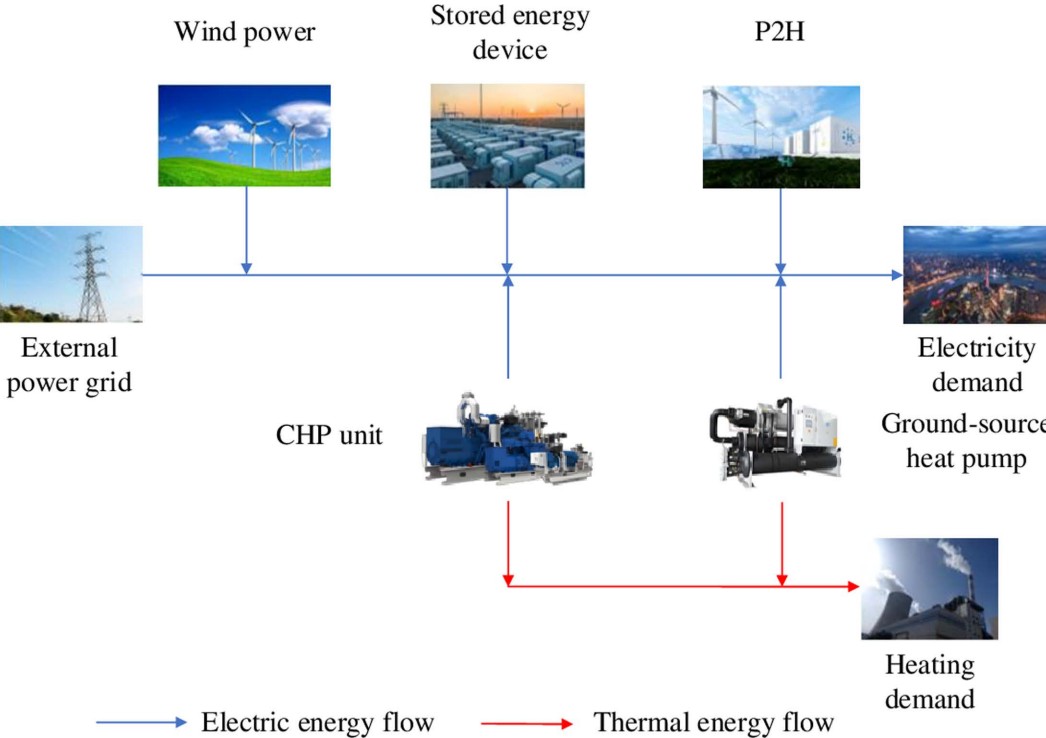

**Fig 1. Structure diagram of electricity heat hydrogen regional integrated energy system.**

especially suitable for heating systems that require a large amount of thermal energy. The electric and thermal output constraints of the back-pressure CHP unit are shown in (1) and (2).

$$P_{\text{CHP,min}} \leq P_{\text{CHP},t} \leq P_{\text{CHP,max}} \tag{1}$$

$$H_{\text{CHP},t} = \omega P_{\text{CHP},t} \tag{2}$$

Where $P_{\text{CHP},t}$ represents the electricity output of the CHP unit in time period $t$. $H_{\text{CHP},t}$ represents the heat output of the CHP unit in time period $t$. $P_{\text{CHP,min}}$ represents the lower limit of electricity output of the CHP unit in each time period. $P_{\text{CHP,max}}$ represents the upper limit of electricity output of the CHP unit in each time period. $\omega$ represents the ratio coefficient between the electricity output and heat output of the CHP unit.

## 2.2. Modelling of P2H device

In normal operating mode, the efficiency of the P2H device is approximately linearly related to the power consumption of the P2H device. It is generally assumed that the efficiency of the P2H device is constant. The model for the relationship between the hydrogen production of the P2H device and the power consumption of the P2H device is represented in (3).

$$V_{\text{H2},t} = \frac{P_{\text{P2H},t} \Delta t}{\mu_{\text{h}}} \tag{3}$$

Where $V_{\text{H2},t}$ represents the hydrogen production volume of the electrolyzer in time period $t$. $P_{\text{P2H},t}$ represents the power consumption for the electrolyzer in time period $t$. $\mu_{\text{h}}$ represents the specific energy consumption of the electrolyzer, generally taken as 4.50–5.04 (kW·h/Nm³).

## 2.3. Modelling of ground-source heat pump

In the electricity heat hydrogen regional integrated energy system, the thermal energy input of the ground-source heat pump is assumed to remain constant and the heat dissipation loss of the ground-source heat pump is not considered. The working characteristics of the ground-source heat pump can be expressed using the heating coefficient of performance (COP). The relationship between the heat output and electricity input of the ground-source heat pump is shown in (4) and (5).

$$H_{\text{HP},t} = COP \cdot P_{\text{HP},t} \tag{4}$$

$$H_{\text{HP,min}} \leq H_{\text{HP},t} \leq H_{\text{HP,max}} \tag{5}$$

Where $H_{\text{HP},t}$ represents the heat output of the ground-source heat pump during time period $t$. $P_{\text{HP},t}$ represents the electricity consumption of the ground-source heat pump during time period $t$. $COP$ represents the heating coefficient of performance of the ground-source heat pump. $H_{\text{HP,min}}$ represents the minimum heat output of the ground-source heat pump and $H_{\text{HP,max}}$ represents the maximum heat output of the ground-source heat pump.

## 2.4. Modelling of energy storage device

Energy storage devices can transfer energy across time periods, achieving peak shaving and valley filling, thereby enhancing the stability and reliability of the system. The mathematical model is shown in (6)–(12).

Charging and discharging power constraints of energy storage devices:

$$0 \leq P_{S,dis,t} \leq U_{S,dis,t} P_{S,max} \tag{6}$$

$$0 \leq P_{S,ch,t} \leq U_{S,ch,t} P_{S,max} \tag{7}$$

$$P_{S,max} = \mu E_S \tag{8}$$

$$0 \leq U_{S,dis,t} + U_{S,ch,t} \leq 1 \tag{9}$$

Where $P_{S,ch,t}$ represents the charging power of the energy storage devices in time period $t$. $P_{S,dis,t}$ represents the discharging power of energy storage devices in time period $t$. $P_{S,max}$ represents the upper limit of the energy storage devices charging and discharging power, $U_{S,ch,t}$ and $U_{S,dis,t}$ respectively represent a binary variable indicating the charging/discharging state of the energy storage devices in time period $t$, 1 indicates charging or discharging, 0 indicates idle. $\mu$ represents the ratio coefficient between the energy storage capacity and the charging and discharging power limit.

Energy storage devices charge and discharge balance constraint:

$$\eta \sum_{t=1}^{N_T} \left[ P_{S,ch,t} \Delta t \right] - \frac{1}{\eta} \sum_{t=1}^{N_T} \left[ P_{S,dis,t} \Delta t \right] = 0 \tag{10}$$

Where $\eta$ represents the efficiency of energy storage devices charge and discharge. This equation ensures that the energy storage maintains a constant energy level at the beginning and end of the operating cycle, facilitating the cyclic scheduling of energy storage.

State of Charge (SOC) constraints of energy storage devices:

$$SOC_t = \left[ E_{S,0} + \eta \sum_{t'=1}^{t} P_{S,ch,t'} \Delta t - \frac{1}{\eta} \sum_{t'=1}^{t} P_{S,dis,t'} \Delta t \right] \bigg/ E_S \tag{11}$$

$$SOC_{min} \leq SOC_t \leq SOC_{max} \tag{12}$$

Where $E_S$ represents the configured capacity of energy storage devices. $E_{S,0}$ represents the initial charge in the energy storage device, set to $0.55E_S$. $SOC_t$ represents the state of charge of energy storage devices in time period $t$. $SOC_{max}$ represents the upper limit of SOC of the energy storage. $SOC_{min}$ represents the lower limit of SOC of the energy storage devices.

## 3. Uncertainty models of supply and demand

In the scenario-based method, typical scenarios consist of representative deterministic wind power and electricity demand curves. However, actual wind power operations exhibit significant randomness and users' electricity demand in daily life is complex and variable. Deviations exist between actual scenarios and typical scenarios. Therefore, in the electricity heat hydrogen regional integrated energy system, uncertainties of wind power output and electricity demand need to be considered. It is essential to account for a certain fluctuation range based on typical scenarios. Using a box-type uncertain set to describe wind power and electricity demand. This set takes typical scenarios as the predicted values. The upper bound of the fluctuation range is represented by the sum of the predicted value and the maximum upward deviation, while the lower bound is represented by the difference between the predicted value and the maximum downward

deviation. The uncertainty set of wind power output and electricity demand in traditional robust optimization can be expressed as the box-type set in (13). This box-type set is independent of the probability distribution of uncertainty parameters, ensuring that the robust optimization results are not affected by the fluctuations parameters of wind power and electricity demand.

$$U = \begin{cases} \boldsymbol{u} = \left[u_{\mathrm{WT},t}, u_{\mathrm{L},t}\right]^{T} \in \mathrm{R}^{N_{\mathrm{T}} \times 2}, t = 1, 2 \cdots N_{\mathrm{T}} \\ u_{\mathrm{WT},t} \in \left[\underline{u_{\mathrm{WT},t}} - \Delta u_{\mathrm{WT},t}^{-}, \underline{u_{\mathrm{WT},t}} + \Delta u_{\mathrm{WT},t}^{+}\right] \\ u_{\mathrm{L},t} = \left[\underline{u_{\mathrm{L},t}} - \Delta u_{\mathrm{L},t}^{-}, \underline{u_{\mathrm{L},t}} + \Delta u_{\mathrm{L},t}^{+}\right] \end{cases} \tag{13}$$

Where U represents the box-type set. $u_{\mathrm{WT},t}$ and $u_{\mathrm{L},t}$ respectively represent the wind power uncertainty variable considering wind power uncertainty and the demand uncertainty variable considering electricity demand uncertainty. $u\underline{\mathrm{WT},t}$ and $u\underline{\mathrm{L},t}$ respectively represent the predicted value of wind power and electricity demand in time period $t$. $N_{\mathrm{T}}$ represent the operating cycle. $\Delta u^{+}_{\mathrm{WT},t}$ and $\Delta u^{-}_{\mathrm{WT},t}$ respectively represent the maximum upper and lower fluctuation deviations of wind power, $\Delta u^{+}_{\mathrm{L},t}$ and $\Delta u^{-}_{\mathrm{L},t}$ respectively represent the maximum upper and lower fluctuation deviations of electricity demand.

The box-type set representation in (13) is used to characterize the uncertainty of wind power and electricity demand variables by using upper and lower bounds to describe the range of fluctuations. However, this method oversimplifies the expression of uncertainty, leading to a broad generalization of variable fluctuation ranges that includes events with extremely low probabilities. In practice, these events may occur almost negligibly. Because the goal of robust optimization is to ensure the reliable operation of the system under all circumstances, the decision strategy becomes overly cautious to address events with very low probabilities, resulting in overly conservative system configuration schemes. To address this problem, this paper introduces an uncertainty adjustment parameter $\Gamma$ in (13) and adopts a more accurate expression of uncertainty. This approach ensures the reliability of the electricity heat hydrogen regional integrated energy system while more reasonably pursuing economical efficiency. By adjusting the parameter $\Gamma$, decision-makers can adjust the total number of periods in which wind power and demand take on the worst-case scenarios within the box-type uncertainty set U. The larger the value of the parameter $\Gamma$, the more time periods correspond to the worst-case scenarios, making the system configuration more conservative and consequently, the economic efficiency deteriorates. Decision-makers can make full use of the uncertainty adjustment parameter $\Gamma$, balancing between reliability and economic efficiency according to the system's needs. In this paper, the lower bound of wind power is taken as the worst-case scenario and the upper bound of electricity demand is taken as the worst-case scenario. The box-type set with the introduction of the uncertainty adjustment parameter $\Gamma$ is shown in (14).

$$U = \begin{cases} \boldsymbol{u} = \left[u_{\mathrm{WT},t}, u_{\mathrm{L},t}\right]^{\mathrm{T}} \in \mathrm{R}^{N_{\mathrm{T}} \times 2}, t = 1, 2 \cdots N_{\mathrm{T}} \\ u_{\mathrm{WT},t} = \overset{\wedge}{u}_{\mathrm{WT},t} - B^{-}_{\mathrm{WT},t} \Delta u^{-}_{\mathrm{WT},t} \\ \sum_{t=1}^{N_{\mathrm{T}}} B^{-}_{\mathrm{WT},t} \leq \Gamma_{\mathrm{WT}}, 0 \leq B^{-}_{\mathrm{WT},t} \leq 1 \\ u_{\mathrm{L}}(t) = \overset{\wedge}{u}_{\mathrm{L},t} + B^{+}_{\mathrm{L},t} \Delta u^{+}_{\mathrm{L},t} \\ \sum_{t=1}^{N_{\mathrm{T}}} B^{+}_{\mathrm{L},t} \leq \Gamma_{\mathrm{L}}, 0 \leq B^{+}_{\mathrm{L},t} \leq 1 \end{cases} \tag{14}$$

Where $B^{-}_{\mathrm{WT},t}$, $B^{+}_{\mathrm{L},t}$ are both binary variables. When its value is 0, the uncertainty variable for the corresponding time period takes the predicted value. When its value is 1, the wind power

uncertainty variable for the corresponding time period takes the lower bound value and the electricity demand uncertainty variable takes the upper bound value. $\Gamma_{WT}$ and $\Gamma_{L}$ are uncertainty adjustment parameters for wind power and electricity demand, respectively, both being integers within the range of 0 and $N_{T}$.

## 4. The two-stage robust optimization configuration of the electricity heat hydrogen regional integrated energy system

### 4.1. Objective function

To fully harness the wind energy of the electricity heat hydrogen regional integrated energy system, a two-stage robust optimization configuration model is established, considering the uncertainties of wind power and electricity demand. This model aims to minimize the sum of investment and operation costs. The investment costs include the annualized costs of wind power, CHP units, ground-source heat pumps, P2H devices and energy storage devices. The operation costs encompass the annualized maintenance costs of each device, electricity purchasing costs, fuel costs and hydrogen sales revenue. The two-stage robust optimization configuration model divides robustness modeling into preoptimization and robust optimization phases. The objective function is shown in (15), with the "min" in the preoptimization phase representing configuration optimization and the "max min" in the robust optimization phase representing operational optimization.

$$\min C = \min_{x}\left(C_{int} + \max_{u}\min_{y} C_{ope}\right) \qquad (15)$$

Where $x$ represents the configuration variable, which is the decision variable in the first stage, specified as in (16). $y$ represents the operational variable, which is the decision variable in the second stage, specified as in (17). $y$ is jointly determined by the configuration variable $x$ and the uncertainty variable $u$. The objective of the "max min" is to find the minimum operating cost of the system under the worst-case scenarios of wind power and electricity demand variations. The outer "max" represents finding the worst-case scenarios of wind power and electricity demand across all $u$, while the inner "min" indicates finding the minimum $y$ given $x$ and $u$. $u$ is the uncertainty set for wind power and electricity demand established in (14). $C_{int}$ represents investment costs and $C_{ope}$ represents operating costs, as specified in (18) and (19), respectively.

$$x = \left[E_{WT}, E_{CHP}, E_{P2H}, E_{HP}, E_{S}\right]^{T} \qquad (16)$$

$$y = \left[P_{WT,t}, P_{CHP,t}, P_{HP,t}, P_{S,ch,t}, P_{S,dis,t}, P_{P2H,t}, P_{buy,t}, P_{L,t}\right]^{T}, t = \left(1,2\cdots N_{T}\right) \qquad (17)$$

Where $E_{WT}$, $E_{CHP}$, $E_{P2H}$, $E_{HP}$, $E_{S}$, represent the configured capacities of wind power, CHP units, P2H devices, ground-source heat pumps and energy storage devices, respectively. $P_{WT,t}$, $P_{buy,t}$, $P_{L,t}$ represent the actual output of wind power, purchased power and demand power for time period $t$, respectively.

Investment costs include the annualized configuration costs of wind power, CHP units, P2H devices, ground-source heat pumps and energy storage devices, as specified in (18).

$$C_{int} = \sum_{i\in I}\frac{\rho(1+\rho)^{r_{i}}}{(1+\rho)^{r_{i}}-1}c_{i}E_{i} \qquad (18)$$

Where $\rho$ represents the discount rate. $i$ represents the type of equipment. I represents the set of equipment types, including wind power, CHP units, P2H devices, ground-source heat pumps and

energy storage devices. $r_i$ represents the discounted years for equipment $i$. $c_i$ represents the unit investment cost for equipment $i$. $E_i$ represents the configured capacity for equipment $i$.

Operating costs include the maintenance costs of equipments, fuel costs and hydrogen sales revenue, as specified in (19).

$$C_{\text{ope}} = C_{\text{om}} + C_{\text{fuel}} + C_{\text{buy}} - C_{\text{H}_2} \tag{19}$$

Where $C_{\text{om}}$, $C_{\text{fuel}}$, $C_{\text{buy}}$ represent the maintenance costs of the equipments, fuel costs, electricity purchasing costs and hydrogen sales revenue, respectively, as specified in (20).

$$\begin{cases} C_{\text{om}} = \sum_{i \in I} \sum_{t=1}^{N_{\text{T}}} c_{i,\text{om}} P_{i,t} \Delta t \\ C_{\text{fuel}} = \sum_{t=1}^{N_{\text{T}}} c_{\text{fuel}} P_{\text{CHP},t} \Delta t \\ C_{\text{buy}} = \sum_{t=1}^{N_{\text{T}}} c_{\text{buy}} P_{\text{buy},t} \Delta t \\ C_{\text{H2}} = \sum_{t=1}^{N_{\text{T}}} c_{\text{H2}} V_{\text{H2},t} \Delta t \end{cases} \tag{20}$$

Where $c_{i,\text{om}}$ represents the unit maintenance cost of equipment $i$. $P_{i,t}$ represents the power of equipment $i$ in time period $t$. $c_{\text{fuel}}$ represents the unit fuel cost for CHP units. $c_{\text{buy}}$ represents the electricity price. $c_{\text{H2}}$ represents the unit selling price of hydrogen.

## 4.2  Constraint condition

The optimization configuration model of the electricity heat hydrogen regional integrated energy system includes constraints such as power balance constraints, heat balance constraints, capacity configuration constraints, equipment constraints and purchased power constraints.

### 4.2.1.  Constraint on heat balance.

$$H_{\text{CHP},t} + H_{\text{HP},t} = H_{\text{L},t} \tag{21}$$

Where $H_{\text{L},t}$ represents the power of the heat demand in time period $t$.

### 4.2.2.  Constraint on power balance.

$$P_{\text{S,dis},t} + P_{\text{CHP},t} + P_{\text{WT},t} + P_{\text{buy},t} = P_{\text{L},t} + P_{\text{S,ch},t} + P_{\text{P2H},t} \tag{22}$$

### 4.2.3.  Constraint on capacity configuration.

$$0 \leq E_i \leq E_{i,\text{max}} \tag{23}$$

Where $E_{i,\text{max}}$ represents the upper limit of the installed capacity of equipment $i$.

### 4.2.4.  Constraint on P2H device power.

$$0 \leq P_{\text{P2H},t} \leq P_{\text{P2H},\text{max}} \tag{24}$$

Where $P_{\text{P2H,max}}$ represents the upper limit of the electrolyzer capacity.

### 4.2.5.  Constraint on purchase power.

$$0 \leq P_{\text{buy},t} \leq P_{\text{buy,max}} \tag{25}$$

Where $P_{\text{buy,max}}$ represents the maximum value of purchased power.

**4.2.6. Constraint on energy storage device.** The constraints on energy storage devices are shown in (6)–(12).

**4.2.7. Constraint on CHP unit.** The constraints on CHP units output power are shown in (1) and (2).

**4.2.8. Constraint on ground-source heat pump.** The constraints on ground-source heat pumps output power are shown in (4) and (5).

## 4.3. Solution method

This paper uses the C&CG algorithm to solve the two-stage robust optimization configuration problem. The C&CG algorithm decomposes the original problem into master and subproblem for alternating solutions, thereby accelerating the solving efficiency. Compared with the traditional Benders algorithm, the C&CG algorithm can complete the convergence in a short time, with fast solving speed and high efficiency. The specific reasons are as follows: (1) The C&CG algorithm can shorten the range of function solving by strictly screening the scene; (2) When calculating the main problem, the C&CG algorithm retains the original model structure of the function, while the theme of Benders' construction is mainly based on the dual cut plane, which destroys the original model structure and complicates the solution process. Therefore, this paper uses the C&CG algorithm to solve the robust optimal configuration model of the electricity heat hydrogen regional integrated energy system. The subproblem is a nonlinear problem and prior to alternating solution, it needs to be linearized using strong duality theory and auxiliary variables. The solution is then obtained by invoking the Cplex solver in Matlab simulation software, ultimately yielding the solution to the original problem.

In order to better present the implementation of the solution algorithm, equations (1)–(25) are written in matrix form, as shown in (26).

$$
\begin{cases}
\min_{x}\left(c^{T}x + \max_{u,y}\min b^{T}y\right) \\
s.t. \quad Dx \geq d \\
\qquad Fx + Gy \geq h \\
\qquad Ky = k \\
\qquad Ey \geq e
\end{cases}
\tag{26}
$$

Where $c$ and $b$ represent coefficient column vectors corresponding to (15). $d$, $h$, $k$, $e$ represent the constant column vector corresponding to the respective constraints. $D$, $F$, $G$, $K$, $E$ represent the coefficient matrix corresponding to the respective constraints.The objective function corresponds to (15). The constraints $Dx \geq d$ corresponds to (23). The constraints $Fx+Gy \geq h$ encompasses (1), (5)–(9), (11), (12), (24). The constraints $Ky=k$ involves (2), (4), (10), (21), (22). The constraints $Ey \geq e$ corresponds to (25).

Decomposing (26) yields the master problem as shown in (27).

$$
\begin{cases}
\min_{x}\left(c^{T}x_{l} + a\right) \\
s.t. \quad a \geq b^{T}y_{l} \\
\qquad Dx_{l} \geq d \\
\qquad Fx_{l} + Gy_{l} \geq h \\
\qquad Ky_{l} = k \\
\qquad Ey_{l} \geq e \\
\qquad Iy_{l} = u_{l}^{*} \\
\qquad \forall l \leq m
\end{cases}
\tag{27}
$$

Where $l$ represents the iteration count. $\boldsymbol{a}$ represents an auxiliary variable. $\boldsymbol{x}_l$ and $\boldsymbol{y}_l$ represent the solutions after $l$ iterations. $\boldsymbol{u}_l^*$ represents the value of $\boldsymbol{u}$ after $l$ iterations. $m$ represents the maximum number of iterations. $\boldsymbol{Iy}_l=\boldsymbol{u}_l^*$ represents the values of wind power and electricity demand obtained in the worst case after $l$ iterations.

Decomposing (26) yields the subproblem shown in (28).

$$\max_{\boldsymbol{u}\in U} \min_{\boldsymbol{y}\in\Omega(\boldsymbol{x},\boldsymbol{u})} \boldsymbol{b}^T\boldsymbol{y}_l \tag{28}$$

Where $\Omega(\boldsymbol{x},\boldsymbol{u})$ represents the feasible set of $\boldsymbol{y}$ given $(\boldsymbol{x},\boldsymbol{u})$, as shown in (29).

$$\Omega(\boldsymbol{x},\boldsymbol{u})=\begin{cases} \boldsymbol{y}_l \mid \\ \boldsymbol{Ey}_l \geq \boldsymbol{e}, & \to \boldsymbol{\gamma} \\ \boldsymbol{Ky}_l = \boldsymbol{k}, & \to \boldsymbol{\lambda} \\ \boldsymbol{Fx}_l + \boldsymbol{Gy}_l \geq \boldsymbol{h}, & \to \boldsymbol{v} \\ \boldsymbol{Iy}_l = \boldsymbol{u}. & \to \boldsymbol{\pi} \end{cases} \tag{29}$$

Where $\boldsymbol{\gamma}, \boldsymbol{\lambda}, \boldsymbol{v}, \boldsymbol{\pi}$ represent the dual variables corresponding to each constraint of the subproblem.

Based on (29) and strong duality theory, the inner "min" in (28) is converted to "max" form and merged with the outer "max", resulting in the dual problem as shown in (30).

$$\begin{cases} \max_{\boldsymbol{u}\in U,\, \boldsymbol{\gamma},\boldsymbol{\lambda},\boldsymbol{v},\,\boldsymbol{\pi}} \boldsymbol{e}^T\boldsymbol{\gamma}+\boldsymbol{k}^T\boldsymbol{\lambda}+(\boldsymbol{h}-\boldsymbol{Fx})^T\boldsymbol{v}+\boldsymbol{u}^T\boldsymbol{\pi} \\ s.t.\ \boldsymbol{E}^T\boldsymbol{\gamma}+\boldsymbol{K}^T\boldsymbol{\lambda}+\boldsymbol{G}^T\boldsymbol{v}+\boldsymbol{I}^T\boldsymbol{\pi}\leq\boldsymbol{c} \\ \boldsymbol{\gamma}\geq0,\ \boldsymbol{\lambda}\geq0,\ \boldsymbol{v}\geq0 \end{cases} \tag{30}$$

After substituting (14) into (30), by introducing relevant constraints and auxiliary variables to linearize it, we can obtain:

$$\begin{cases} \max_{B,B',\gamma,\lambda,v,\pi} \mathrm{e}^T\gamma+\mathrm{k}^T\lambda+(\mathrm{h}-\mathrm{Fx})^T\mathrm{v}+\hat{u}^T\pi+\Delta u^T B' \\ \\ s.t\,\gamma+\mathrm{K}^T\lambda+\mathrm{G}^T\mathrm{v}+\mathrm{I}^T\pi\leq\mathrm{c} \\ \\ 0\leq B'\leq\bar{\pi}\mathrm{B} \\ \\ \pi-\bar{\pi}(1-\mathrm{B})\leq B'\leq\pi \\ \\ \sum_{t=1}^{N_r}B_{\mathrm{WT},t}\leq\Gamma_{\mathrm{WT}} \\ \\ \sum_{t=1}^{N_r}B_{\mathrm{L},t}\leq\Gamma_{\mathrm{L}} \end{cases} \tag{31}$$

Where $\Delta\boldsymbol{u}=[\Delta u^-_{\mathrm{WT},t}, \Delta u^+_{\mathrm{WT},t}]^T$, $\boldsymbol{B}=[B^-_{\mathrm{WT},t}, B^+_{\mathrm{WT},t}]^T$. $\bar{\boldsymbol{\pi}}$ is the upper bound of the dual variable, which can be taken as a sufficiently large positive real number; $\boldsymbol{B'}=[B'_{\mathrm{WT},t}, B'_{\mathrm{WT},t}]^T$ is the introduced continuous auxiliary variable.

After the above decomposition, the specific process of solving using the C&CG algorithm is as follows:

(1) Initial setup: Set the lower and upper bounds of the objective function values as $L_B = -\infty$ and $U_B = +\infty$, respectively. Also, provide an initial value of uncertain variables $\boldsymbol{u}$, uncertainty adjustment parameter $\Gamma$ and iteration count $l = 0$.

(2) Solution to the master problem: Based on the worst-case scenario $\boldsymbol{u}_1^*$, solve (27) to obtain the optimal solution of the main problem $(\boldsymbol{x}_l^*, \boldsymbol{a}_l^*, \boldsymbol{y}_1^*, \ldots, \boldsymbol{y}_l^*)$ and update $L_B = \boldsymbol{c}^T\boldsymbol{x}_l^* + \boldsymbol{a}_l^*$. Substitute $\boldsymbol{x}_l^*$ into (31).

(3) Solution to the subproblem: Solve (31) to obtain the objective function value $\boldsymbol{b}^T\boldsymbol{y}_l^*$ of the subproblem and the corresponding values of uncertain variables $\boldsymbol{u}_{l+1}^*$ and update $U_B = \min\{U_B, \boldsymbol{c}^T\boldsymbol{x}_l^* + \boldsymbol{b}^T\boldsymbol{y}_l^*\}$.

(4) Convergence criterion: Set the convergence threshold of the algorithm as $\varepsilon$. If $U_B - L_B \leq \varepsilon$, stop iterating and return the optimal solution $\boldsymbol{x}_l^*$ and $\boldsymbol{y}_l^*$; otherwise, increase the variable $\boldsymbol{y}_{l+1}$ and add constraints as in (32).

$$\begin{cases} \boldsymbol{\alpha} \geq \mathbf{b}^T \mathbf{y}_{l+1} \\ \boldsymbol{E}\boldsymbol{y}_{l+1} \geq \boldsymbol{e} \\ \boldsymbol{K}\boldsymbol{y}_{l+1} = \boldsymbol{k} \\ \boldsymbol{F}\boldsymbol{x} + \boldsymbol{G}\boldsymbol{y}_{l+1} \geq \mathrm{h} \\ \boldsymbol{I}\boldsymbol{y}_{l+1} = \boldsymbol{u}_{l+1}^* \end{cases} \tag{32}$$

Let $l = l + 1$ and proceed to step (2), repeating the process until the algorithm converges.

The flowchart for model solving process is shown in Fig 2.

## 5. Case study analysis

### 5.1. Basic data

This paper takes an integrated energy system in the northern part of China as the research object during the heating season. The region has abundant wind and geothermal resources. The maximum allowable capacity for wind power is $5 \times 10^7$ kW and for ground-source heat pumps, it is $4.5 \times 10^6$ kW. The capacity limits for other devices, such as CHP units and energy storage devices, are set at $2.5 \times 10^7$ kW and $1.5 \times 10^7$ kW·h, respectively. There is no installation capacity limit for the P2H devices. The minimum power for ground-source heat pumps, P2H devices and energy storage devices is all set to 0. The system parameters are detailed in Table 1 [13] and the selling price of hydrogen is 3.6 ¥/Nm³ [14]. With a penalty cost for loss of power supply of 10 ¥/kW·h [15].

The typical wind power scenario curve of the electricity heat hydrogen regional integrated energy system during the heating season is shown in Fig 3. The typical scenario curve of electricity demand during the heating season is shown in Fig 4. The probabilities of each typical scenario are shown in Tables 2 and 3. The predicted curve for the heating demand during the heating season is shown in Fig 5. Without considering the uncertainty of the heating demand.

The fluctuation deviations of electricity demand and wind power output are generally 10% and 15% of the predicted values, respectively [16]. The uncertainty set of wind power typical day 1, considering fluctuation deviations in typical scenarios, is shown in Fig 6. Similarly, the uncertainty set of electricity demand typical day 1, accounting for fluctuation deviations in typical scenarios, is depicted in Fig 7. The remaining typical days are similar and are not further elaborated here.

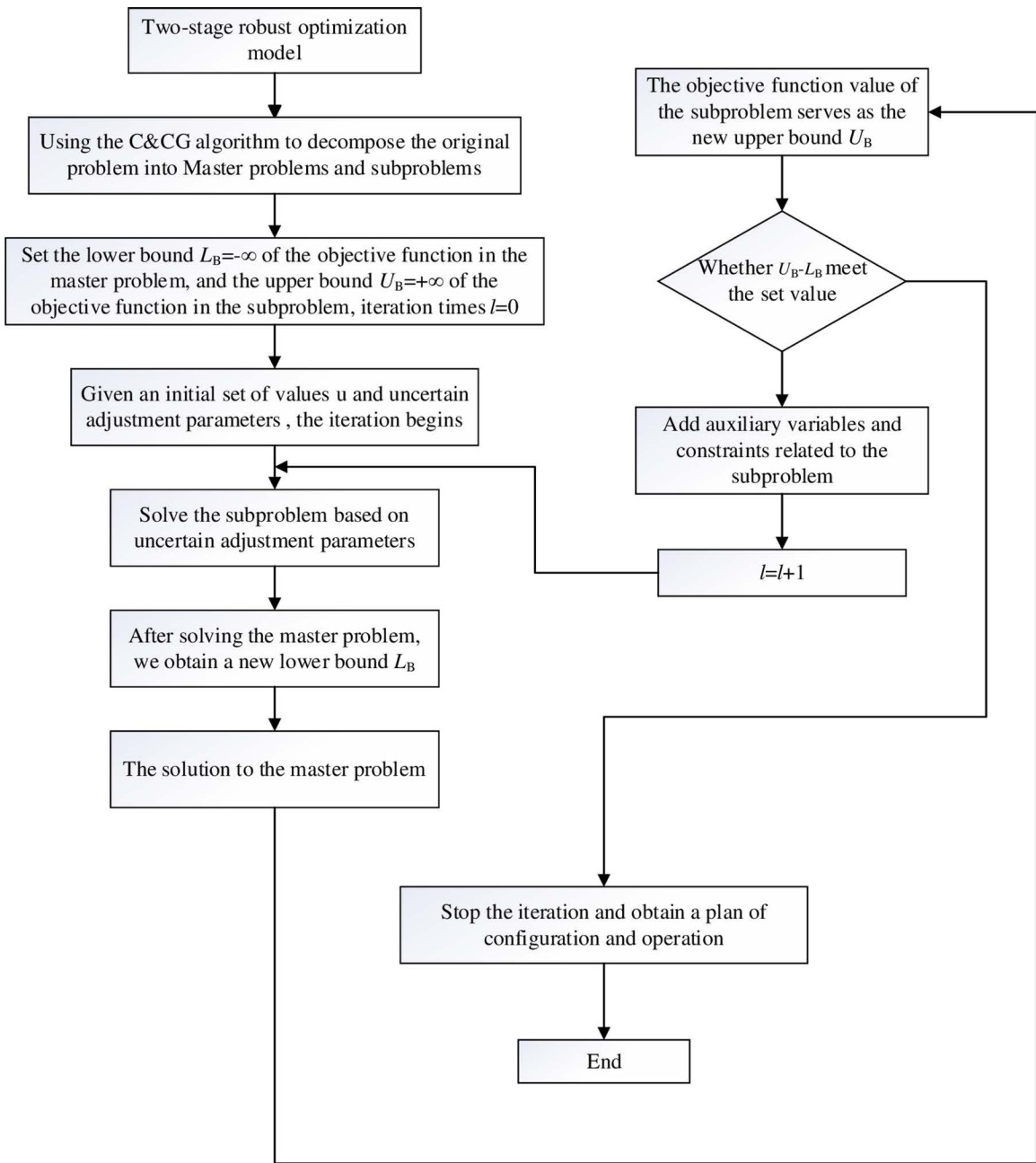

**Fig 2. The flowchart for model solving process.**

## 5.2. Configuration results analysis

Three configuration schemes are established to validate the effectiveness of the proposed model and the reliability of the algorithm in this paper. (1) Robust optimization configuration of the electricity heat hydrogen regional integrated energy system ($\Gamma_{WT} = 13$, $\Gamma_L = 13$). (2) Robust optimization configuration of the electricity heat hydrogen regional integrated energy system ($\Gamma_{WT} = 13$, $\Gamma_L = 13$). (3) Traditional robust optimization configuration of the electricity heat hydrogen regional integrated energy system ($\Gamma_{WT} = 24$, $\Gamma_L = 24$).

**Table 1. Relevant parameters of the electricity heat hydrogen regional integrated energy system.**

| Device type | Parameter | Value | Device type | Parameter | Value |
|---|---|---|---|---|---|
| CHP unit | $c_{CHP}$/¥/kW | 5800 | P2H device | $c_{P2H}$/¥/kW | 2500 |
| | $\omega$ | 0.75 | | $r_{P2H}$/year | 10 |
| | $c_{fuel}$/¥/kW | 0.6 | | $\mu_h$ | 4.7 |
| | $c_{CHP,om}$/¥/kW | 0.059 | | $c_{P2H,om}$/¥/kW | 0.026 |
| | $r_{CHP}$/year | 10 | ground-source heat pump | $c_{HP}$/¥/kW | 3 000 |
| | $P_{CHP,min}$/kW | 20%$E_{CHP}$ | | $c_{HP,om}$/¥/kW | 0.023 |
| energy storage device | $c_S$/¥/kW·h | 5000 | | $r_{HP}$/year | 15 |
| | $\eta$ | 0.95 | | COP | 3.5 |
| | $\mu$ | 0.21 | Wind turbin | $c_{WT}$/¥/kW | 4912 |
| | $SOC_{min}$ | 0.2 | | $c_{WT,om}$/¥/kW | 0.0196 |
| | $SOC_{max}$ | 0.9 | | $r_{WT}$/year | 10 |
| | $r_S$/year | 10 | maximum purchased power | $P_{buy,max}$/$10^4$ kW | 250 |
| | $c_{S,om}$/¥/kW·h | 0.25 | | | |
| discount rate | $\rho$ | 8% | electricity price | $c_{buy}$/¥/kW·h | 0.9 |

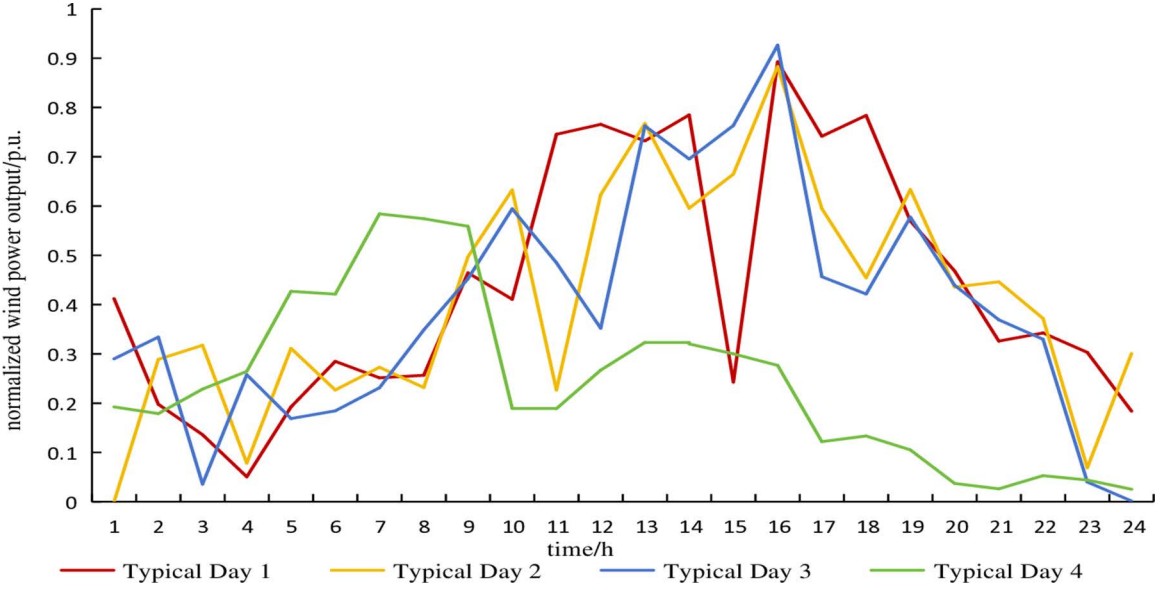

**Fig 3. Typical scenarios of wind power.**

The equipment capacities and costs for each configuration scheme are shown in Table 4. Scheme 1 represents the proposed configuration result of the electricity heat hydrogen regional integrated energy system in this paper. The total cost includes investment and operation costs. The operation cost is the annualized operation cost during the normal winter heating season, covering fuel costs, equipment operation and maintenance costs and hydrogen production revenue. The cost is 82.92 billion ¥, with an annualized investment cost of 51.788 billion ¥, resulting in a total cost of 134.708 billion ¥. In Scheme 2, the P2H device is not included, leading to curtailed wind energy production despite meeting electricity and heating demand. The total cost increased by 0.232 billion ¥ compared to Scheme 1.Scheme 3 represents the results of traditional robust configuration. Due to worse wind power output

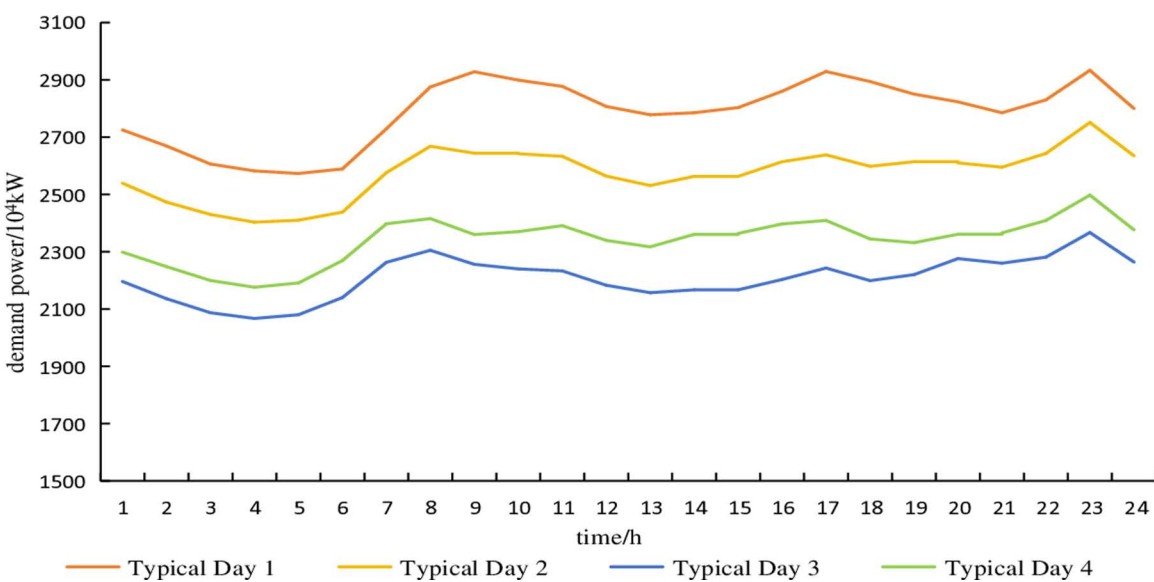

**Fig 4. Typical scenarios of electrical demand.**

**Table 2. Typical daily occurrence probability of wind power.**

| Typical day | Probability |
|---|---|
| 1 | 0.3204 |
| 2 | 0.2361 |
| 3 | 0.1923 |
| 4 | 0.2512 |

**Table 3. Typical daily occurrence probability of electrical demand.**

| Typical day | Probability |
|---|---|
| 1 | 0.57 |
| 2 | 0.23 |
| 3 | 0.06 |
| 4 | 0.14 |

and electricity demand considerations compared to Scheme 1, the storage device capacity, ground-source heat pump capacity, wind power capacity and CHP unit capacity have all increased. Among them, wind power capacity increased the most, by $3.9 \times 10^6$ kW. Conversely, the P2H device significantly decreased by $6.77 \times 10^6$ kW and hydrogen production revenue dropped from 3.074 billion ¥ in Scheme 1 to 0.292 billion ¥. The total cost increased by 2.03% compared to Scheme 1, reaching 2.739 billion ¥.

## 5.3. Analysis of the impact of uncertainty adjustment parameters on capacity configuration

To verify that the uncertainty adjustment parameter $\Gamma$ can flexibly adjust the conservatism of the configuration scheme, the uncertainty adjustment parameter $\Gamma$ is sequentially set from 0 to 24. Here, 0 represents $\Gamma_{WT} = 0$, $\Gamma_L = 0$, 1 represents $\Gamma_{WT} = 1$, $\Gamma_L = 1$ and so on. When $\Gamma_{WT} = 0$,

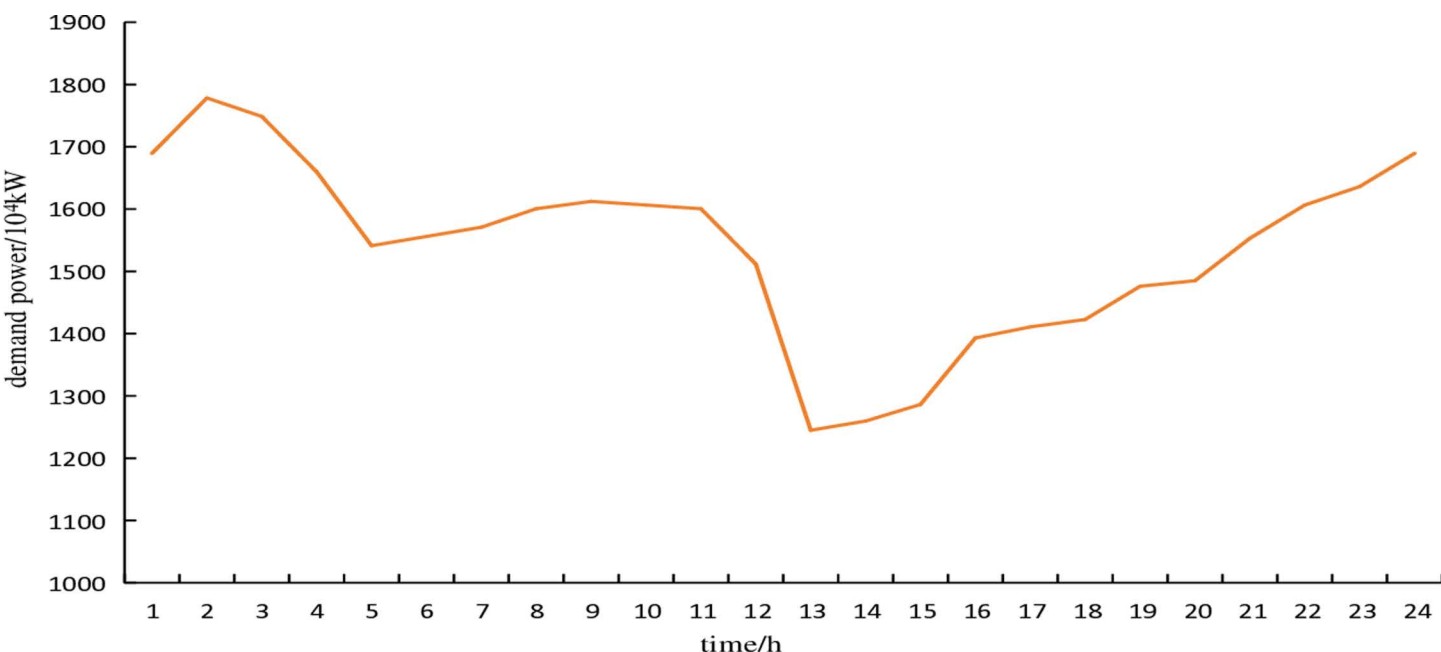

**Fig 5. Heat demand forecast curve.**

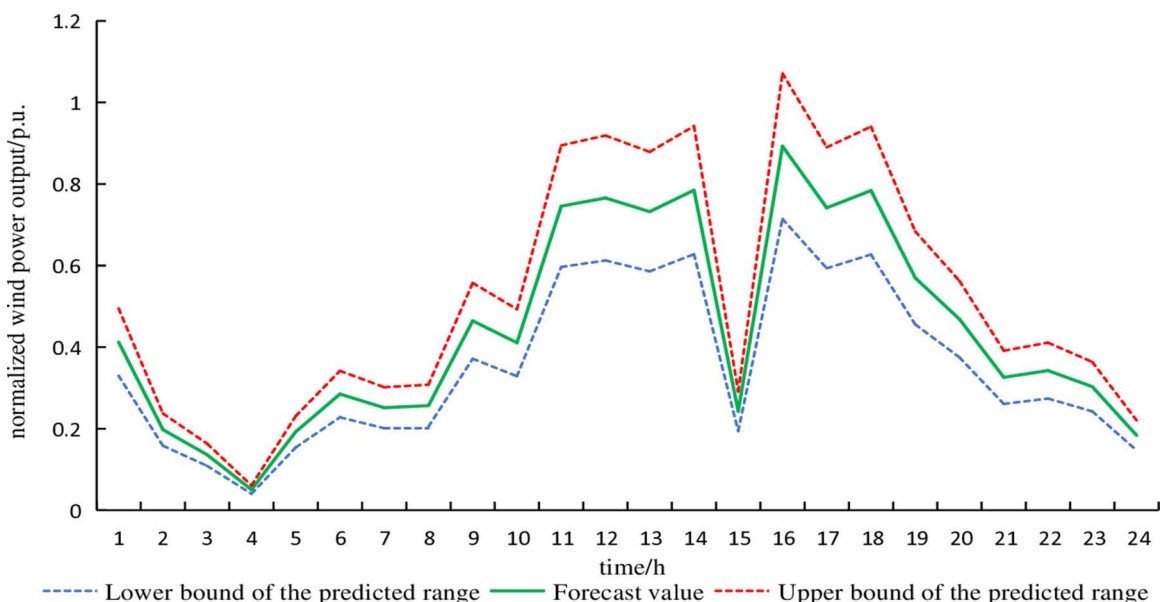

**Fig 6. Forecast curve of wind power on typical day 1.**

$\Gamma_L = 0$, it corresponds to deterministic optimization and when $\Gamma_{WT} = 24$, $\Gamma_L = 24$, it corresponds to traditional robust optimization. The loss of power supply probability (LPSP) is used as the reliability assessment index for the integrated energy system of electric, thermal and hydrogen regions [17]. A smaller LPSP indicates higher reliability of the integrated energy system.

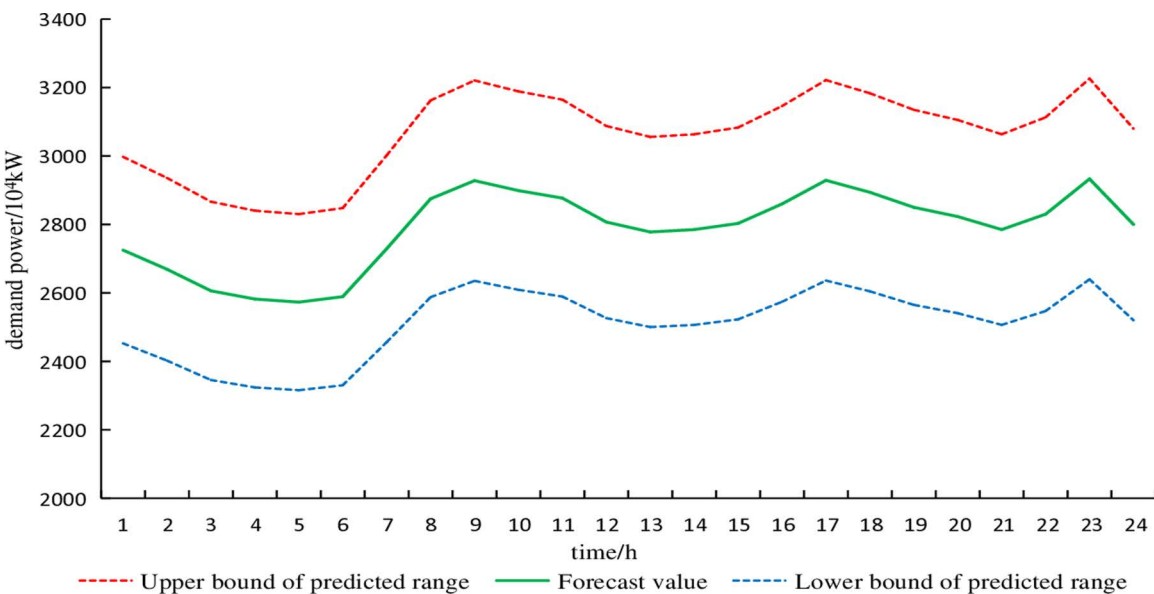

**Fig 7. Forecast curve of electrical demand on typical day 1.**

**Table 4. Equipment capacity and cost for each configuration scheme.**

| Parameter | Scheme 1 | Scheme 2 | Scheme 3 |
|---|---|---|---|
| energy storage device capacity/$10^4$ kW·h | 1068 | 1104 | 1141 |
| ground-source heat pump capacity/$10^4$ kW | 301 | 321 | 324 |
| wind power/$10^4$ kW | 4160 | 4108 | 4550 |
| CHP unit/$10^4$ kW | 2206 | 2206 | 2372 |
| P2H device/$10^4$ kW | 777 | 0 | 100 |
| annual investment cost/$10^9$ ¥ | 517.88 | 489.72 | 532.77 |
| hydrogen production revenue/$10^9$ ¥ | 30.74 | 0 | 2.92 |
| annual operating cost/$10^9$ ¥ | 829.2 | 859.68 | 841.7 |
| wind curtailment rate/% | 0 | 1.45 | 0 |
| total cost/$10^9$ ¥ | 1347.08 | 1349.4 | 1374.47 |

We incorporate the actual measured data of wind power and demand during the winter heating period of the electricity heat hydrogen regional integrated energy system into the robust optimization configuration scheme under various uncertainty adjustment parameters, obtaining the corresponding LPSP. The total cost of this section includes investment costs, fuel costs, operation and maintenance costs, hydrogen sales revenue and loss of power supply penalty costs. The setting of $\Gamma$ and the total costs and LPSP under various uncertainty adjustment parameters $\Gamma$ is shown in Fig 8.

From Fig 8, it can be observed that when $\Gamma_{WT} = 0$, $\Gamma_L = 0$, which represents deterministic optimization, the total cost is minimized, but the LPSP is highest, indicating the lowest reliability. As the uncertainty adjustment parameter $\Gamma$ increases, the number of time periods of wind power and electricity demand reach the boundary of the interval increases, leading to a gradual increase in total cost and a gradual decrease in LPSP. This indicates that the more extreme scenarios of wind power and electricity demand are considered during system planning and configuration, the more conservative the configuration scheme becomes. This

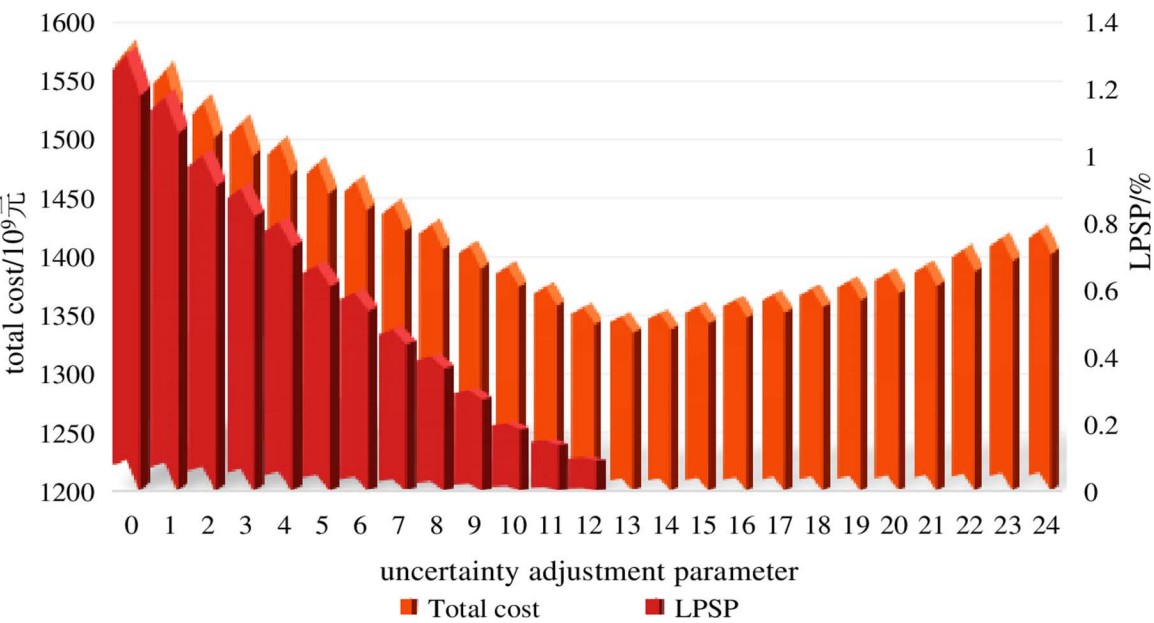

**Fig 8. LPSP and total cost under different uncertainty adjustment parameters.**

conservatism is reflected in the configuration scheme adopting more measures to deal with extreme situations in response to the adverse effects of uncertainty, ensuring stable operation of the system in the face of uncertainty. Conservative configuration schemes typically come with increased total costs but guarantee power supply reliability.

The capacity of each device under different uncertainty adjustment parameters is shown in Figs 9 and 10.

From Figs 9 and 10, it can be observed that the capacities of wind power and CHP units are relatively high, while the capacities of energy storage device, ground source heat pump, and P2H device are relatively low. Additionally, except for the P2H device, the capacities of other equipment increase with the rise of uncertainty parameters.

The variation of investment costs with uncertainty parameters is shown in Fig 11. A comprehensive analysis with Figs 9 and 10 reveals the following: when the uncertainty parameter $\Gamma = 0$, representing deterministic optimization, wind power and electric load remain stable and equal to the predicted values. In this case, the installed capacities of all system equipment, except for P2H devices, are minimal, resulting in the lowest investment cost. As the uncertainty adjustment parameter increases, the number of periods where wind power and electric load reach the interval boundaries rises, considering more uncertainties. Consequently, the storage capacity, ground source heat pump capacity, wind power capacity, and CHP unit capacity of the integrated electricity-heat-hydrogen energy system all increase, while the configured P2H device capacity decreases correspondingly, leading to a gradual rise in investment costs.

Although the reliability of the system increases with the increase of the uncertainty adjustment parameter $\Gamma$, when $\Gamma$ reaches a certain level, the effectiveness of further increasing $\Gamma$ to enhance reliability gradually diminishes. When $\Gamma$ is at its maximum value of 24 (traditional robust optimization), the result becomes overly conservative, leading to unnecessary costs and capacity waste. It is essential to choose the uncertainty adjustment parameter $\Gamma$ reasonably to make robust optimization both reliable and economical, aligning better with practical scenarios. In this case study, when $\Gamma = 13$, the LPSP is 0, indicating that the system meets reliability

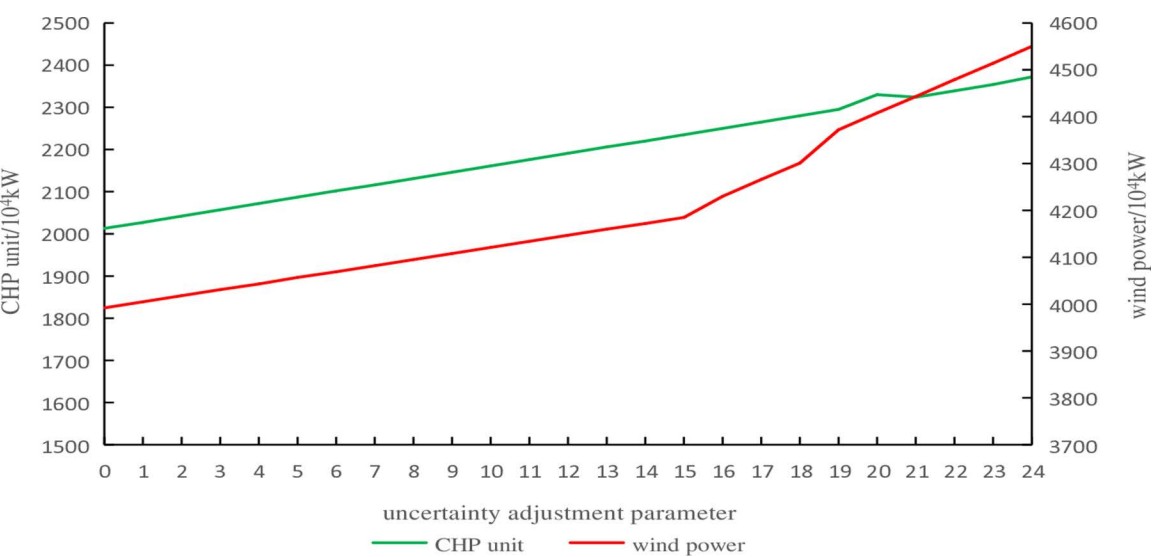

**Fig 9.  The capacities of CHP units and wind power under different uncertainty adjustment parameters.**

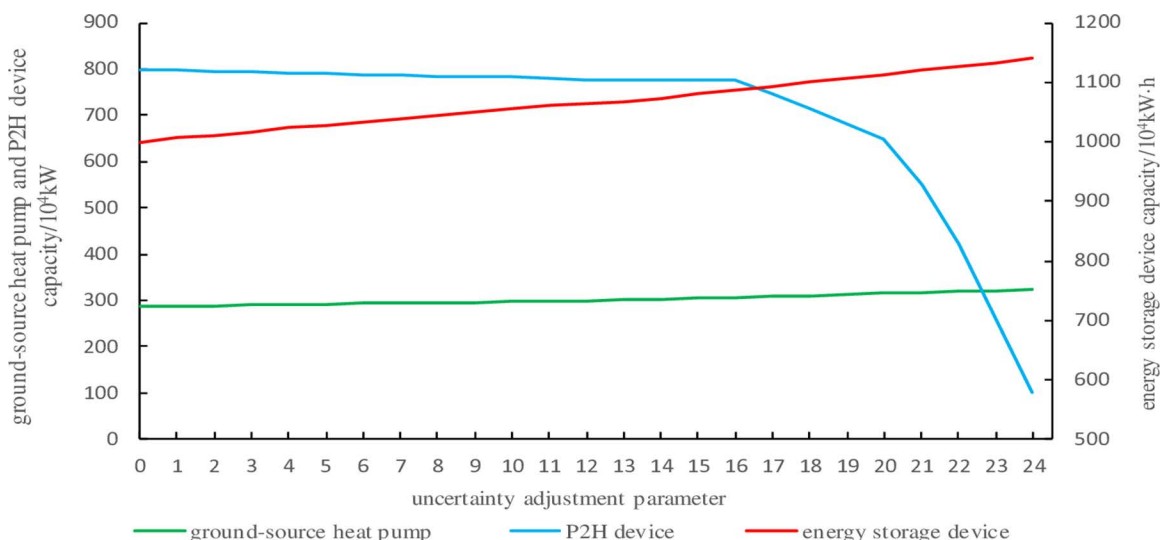

**Fig 10.  The capacities of ground source heat pumps, P2H equipment, and energy storage devices under different uncertainty adjustment parameters.**

requirements. Further increasing $\Gamma$ would lead to an uneconomical configuration. Therefore, when formulating the configuration scheme for the electricity heat hydrogen regional integrated energy system, we need to prioritize economic requirements while ensuring reliability requirements are met.

### 5.4.  Comparative analysis of operational reliability

Two models are set up for comparison to verify the reliability of the configuration results obtained using the methods in the paper. (1) Deterministic optimization configuration model

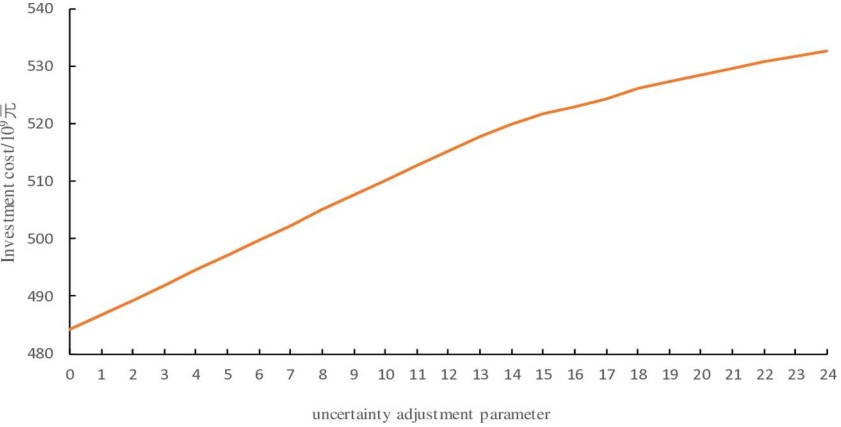

**Fig 11. Investment costs under different uncertainty adjustment parameters.**

for the electricity heat hydrogen regional integrated energy system. (2) Robust optimization configuration model for the electricity heat hydrogen regional integrated energy system considering supply-demand uncertainties ($\Gamma_{WT} = 13$, $\Gamma_L = 13$).

To evaluate the reliability of the system, three worst-case scenarios are selected with wind power and electricity demand prediction errors of 5%, 10% and 15% on a typical day 1 and these worst-case scenarios are applied to the configuration results of the two different models for operation. These three prediction errors represent the degree to which the predicted values of wind power and electricity demand deviate from the actual values, corresponding to wind power actual values of 95%, 90% and 85% and electricity demand actual values of 105%, 110% and 115%, respectively. The configuration details and operational costs are shown in Table 5. The operation of the P2H unit, system purchasing power and loss of power supply situations for the two models under each prediction error are shown in Figs 12–17.

In Table 5, the differences between the robust optimization configuration model and the deterministic optimization configuration model are clearly evident. Specifically, compared to the deterministic model, the robust model shows an increase in energy storage devices capacity, ground-source heat pump capacity, wind power capacity and CHP unit capacity by 6.8%,

**Table 5. Configuration results of each model and the corresponding costs under different prediction errors.**

| Parameter | Model 1 | | | Model 2 | | |
|---|---|---|---|---|---|---|
| Prediction error | 5% | 10% | 15% | 5% | 10% | 15% |
| Energy storage device capacity/$10^4$ kW·h | 1000 | 1000 | 1000 | 1068 | 1068 | 1068 |
| Ground-source heat Pump capacity/$10^4$ kW | 287 | 287 | 287 | 301 | 301 | 301 |
| Wind power/$10^4$ kW | 3992 | 3992 | 3992 | 4160 | 4160 | 4160 |
| CHP unit/$10^4$ kW | 2013 | 2013 | 2013 | 2206 | 2206 | 2206 |
| P2H device/$10^4$ kW | 799 | 799 | 799 | 777 | 777 | 777 |
| Annual investment cost/$10^9$ ¥ | 488.06 | 488.06 | 488.06 | 517.88 | 517.88 | 517.88 |
| Annual operating cost/$10^9$ ¥ | 827.2 | 920.16 | 1091.05 | 800.95 | 923.96 | 1008.75 |
| Annual purchasing cost/$10^9$ ¥ | 15.8 | 43.69 | 75.68 | 0 | 0.19 | 36.77 |
| LPSP | 0 | 0.4 | 1.42 | 0 | 0 | 0 |
| Hydrogen production revenue/$10^9$ ¥ | 7.60 | 0 | 0 | 11.19 | 0 | 0 |
| Penalty cost of loss of power supply/$10^9$ ¥ | 0 | 106.24 | 398.53 | 0 | 0 | 0 |
| Total cost/$10^9$ ¥ | 1331.06 | 1558.15 | 2053.32 | 1318.83 | 1442.03 | 1563.4 |

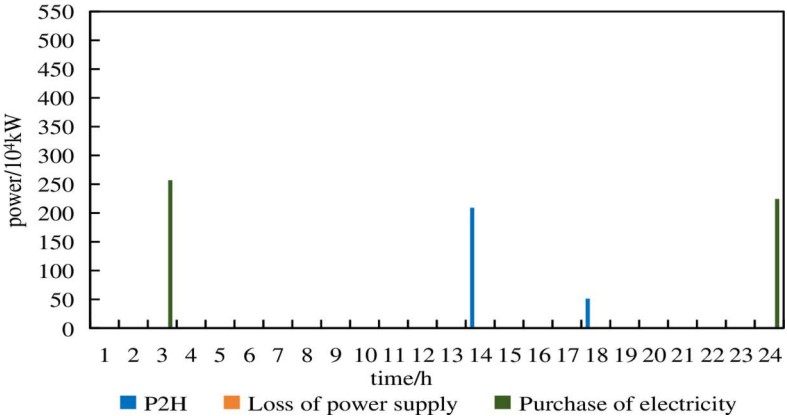

**Fig 12. P2H, electricity purchase and power shortage of Model 1 under 5% prediction errors.**

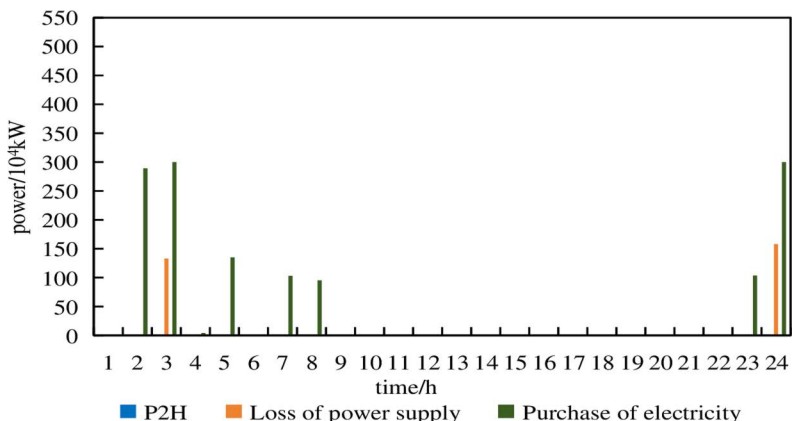

**Fig 13. P2H, electricity purchase and power shortage of Model 1 under 10% prediction errors.**

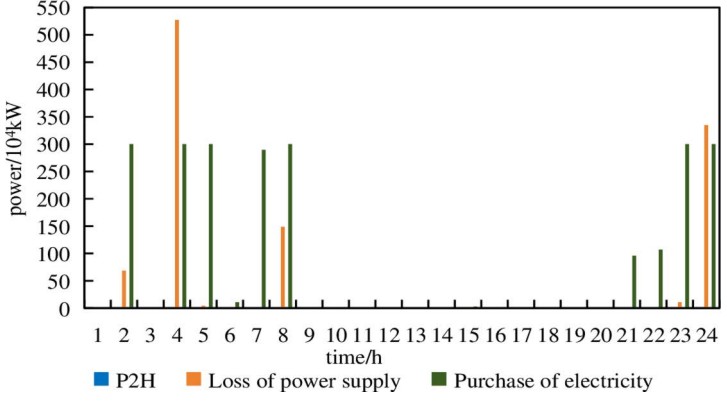

**Fig 14. P2H, electricity purchase and power shortage of Model 1 under 15% prediction errors.**

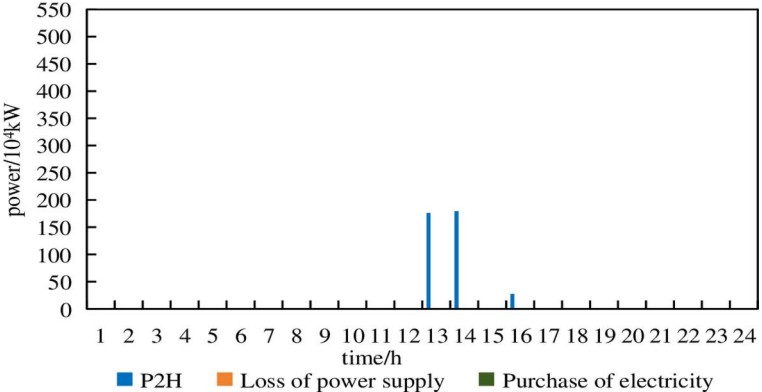

**Fig 15. P2H, electricity purchase and power shortage of Model 2 under 5% prediction errors.**

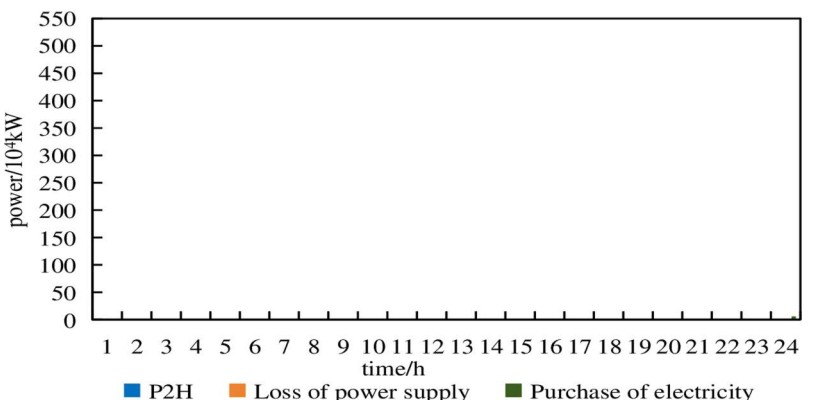

**Fig 16. P2H, electricity purchase and power shortage of Model 2 under 10% prediction errors.**

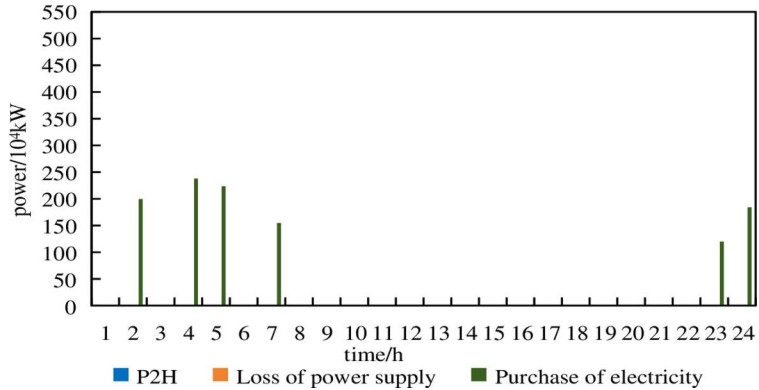

**Fig 17. P2H, electricity purchase and power shortage of Model 2 under 15% prediction errors.**

0.49%, 4.21% and 9.59%, respectively, while the capacity of the P2H unit decreases by 2.75%. The total investment cost also shows an upward trend, increasing by 6.11%. These differences primarily stem from the fact that the equipment capacities configured by the deterministic

optimization model just meet the demand requirements, without fully considering the uncertainty of wind power and electricity demand. However, this also makes the configuration scheme of this model relatively fragile in dealing with the fluctuations of wind power and electricity demand. In contrast, the robust optimization configuration model takes into account the uncertainty of wind power and electricity demand more comprehensively, ensuring that the configured equipment capacities meet the supply-demand balance within the range of wind power and electricity demand fluctuations. Therefore, in this model, except for the P2H device, the capacities of other equipment are more abundant compared to the deterministic optimization configuration model.

At the same time, it can be observed that under the same prediction error conditions, the robust optimization configuration model exhibits lower loss of power supply penalties, lower purchasing costs and lower total costs compared to the deterministic optimization configuration model. Additionally, the hydrogen production revenue is higher in the robust optimization configuration model than in the deterministic optimization configuration model. When the prediction error is 5%, the purchasing rate of the robust optimization configuration model is 0. When the prediction error is 10%, it is almost 0. And when the prediction error is 15%, it is only 1.45%. In contrast, the deterministic optimization configuration model has a purchasing rate of 0.68% when the prediction error is 5%, 1.81% when the prediction error is 10% and 2.99% when the prediction error is 15%. This indicates that the robust optimization configuration model has a lower dependency on the external power grid when there are prediction errors in wind power and electricity demand. As the prediction error increases, apart from the decrease in hydrogen production revenue, the rest of the costs for both models gradually increase. The gap between the two configuration models in terms of loss of power supply probability and total cost widens as the prediction error increases, while the economic advantage of the robust optimization configuration model becomes more prominent. These data clearly demonstrate the high reliability of the configuration scheme of the robust optimization configuration model when facing prediction errors.

From Fig 12, it can be observed that in Model 1, under a prediction error of 5%, electricity purchases occur at 3:00 and 24:00. At 14:00 and 18:00, the geothermal heat pump and energy storage are unable to absorb excess wind power. The remaining excess wind power is converted into hydrogen energy by the P2H device for consumption.

From Fig 13, it can be observed that in Model 1, under a prediction error of 10%, electricity purchases occur during the periods of 2:00–5:00, 7:00–8:00 and 23:00–24:00. Compared to the scenario with a 5% prediction error, the number of purchasing periods and the amount of purchased electricity significantly increase. Additionally, there are instances of loss of power supply at 3:00 and 24:00, which severely impacts the operational safety of the system. The P2H device no longer operates.

From Fig 14, it can be seen that in Model 1, under a prediction error of 15%, electricity purchases occur at 2:00, 4:00–8:00 and 21:00–24:00. Loss of power supply occurs at 2:00, 4:00–5:00, 8:00 and 23:00–24:00. At 4:00, the Loss of power supply amounted to 17.76% of the demand in that period. The P2H device is not operational.

From Fig 15, it can be observed that in Model 2, there are no electricity purchases under a 5% prediction error. The P2H device absorbed excess wind power at 13:00–14:00 and 16:00. Compared to Model 1 under a 5% prediction error, Model 2 performed well.

From Fig 16, it can be seen that in Model 2, under a 10% prediction error, there are only minimal electricity purchases. The system still maintains supply-demand balance, ensuring reliable operation, although the P2H device remains inactive.

From Fig 17, it can be seen that in Model 2, under a 15% prediction error, electricity purchases occur at 2:00, 4:00,5:00,7:00 and 23:00–24:00. Yet there are no instances of power

shortage. This indicates that the robust optimization configuration model exhibits strong power supply reliability by considering uncertainties. In summary, the robust optimization configuration model in this paper balances reliability and economy, achieving the optimal trade-off between economic efficiency and safety.

## 6. Conclusion

To address the problems of wind power curtailment and operational reliability caused by the uncertainty of wind power and electricity demand and to improve the rationality of system capacity configuration, this paper conducts research on capacity optimization configuration of the electricity heat hydrogen regional integrated energy system in the context of dual-side uncertainties of supply and demand. The specific conclusions are as follows:

(1) Based on typical scenarios, a box-type uncertainty set independent of probability distribution is proposed to describe the uncertainty of wind power and demand. By introducing uncertainty adjustment parameters, the reliability of the system is not only enhanced, but also flexible optimization space is provided for the economy of the system. With the adjustment of uncertainty adjustment parameters, the system can adaptively improve the reliability of power supply in different scenarios, especially in the case of large fluctuations in wind power and load, the system can ensure the stability of power supply and effectively reduce economic costs. The flexibility of this method allows it to adapt to a variety of uncertain situations and optimize decision outcomes.

(2) A two-stage robust optimization method considering the uncertainty of source load is proposed, and a min-max-min two-stage robust optimization multi-vector configuration model aiming at the minimum comprehensive system cost is established to optimize the unified configuration of wind power, CHP units, ground source heat pumps, P2H devices, energy storage and other equipment. Through the two-stage robust optimization method, the configuration of the system can not only adapt to the dual-side uncertainties of supply and demand., but also effectively reduce the impact of uncertainty on the operation of the integrated energy system. Compared with the deterministic optimization configuration model, the robust optimization configuration model achieves lower electricity purchase rates and LPSP in actual operation, demonstrating higher reliability.

(3) Models for CHP units, ground-source heat pumps, P2H units and energy storage devices are established. P2H units are jointly optimized with CHP units, wind power, ground-source heat pumps and energy storage devices in the integrated energy system to improve energy utilization efficiency, which can realize the complementarity and coordination of resources, reduce energy waste when the load of other equipment is excessive, improve the efficiency of equipment use, avoid equipment idling, and further reduce the overall operating cost. In the case of excess wind power, the P2H device can effectively convert wind curtailment into hydrogen energy, improve the efficiency of wind energy utilization and decrease wind curtailment rates.

China's 14th Five-Year Plan emphasizes the green transition of the energy structure, promoting the development of low-carbon energy and encouraging the use of clean energy such as wind, solar, and hydrogen, while reducing reliance on traditional fossil fuels. The dual-carbon goals (carbon peaking and carbon neutrality) are particularly significant for northern China. This region has long depended on traditional energy sources like coal, making the transition to a greener energy structure a significant challenge.

In recent years, northern China's energy market has undergone substantial transformation in the development of renewable energy and a low-carbon economy. Air pollution and greenhouse gas emissions caused by coal combustion have become key drivers of

energy structure reform in the region. During the winter heating season, the high demand for heating makes coal-fired heating the primary source of pollution and carbon emissions. Although the government has encouraged the development of green energy, coal still occupies a significant position in the supply and demand dynamics of the energy market.

The electricity heat hydrogen regional integrated energy system optimization method proposed in this paper, takes northern China as a reference. It effectively enhances the integration and utilization of wind power, reduces wind curtailment, and minimizes the consumption of fossil fuels while enabling hydrogen production from curtailed wind power. By ensuring both economic viability and supply reliability, this method aligns well with the dual-carbon goals and provides a viable pathway for achieving them.

On the basis of the existing research, this paper studies the capacity optimization of the integrated energy system in the electrothermal hydrogen region, but there are still some shortcomings. In the following aspects can also be further studied:

(1) The comprehensive energy system of electric hydrogen region has various forms of energy, and model establishment is the primary problem. The heat network model has the characteristics of time lag, so a more accurate mathematical model needs to be established, and the impact of time lag on heating can be further studied in the future.

(2) This paper only optimizes the equipment capacity of the integrated energy system in the electrothermal hydrogen region, without considering the flexibility of each equipment during the operation of the system. How to further combine the actual energy use scenario and add flexibility constraints into the model will be the next research focus.

## Supporting information

**S1 Data. Minimal data set.**
(XLSX)

## Author contributions

**Conceptualization:** Tianhe Sun, Xiaoyi Qian.

**Funding acquisition:** Baoshi Wang.

**Resources:** Xiaoyi Qian.

**Writing – original draft:** Runqi Sun.

**Writing – review & editing:** Tianhe Sun.

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
