## [Decision Letter · Decision Letter 0]

23 Dec 2024

PONE-D-24-55323Capacity Configuration Optimization of Electricity Heat Hydrogen Regional Integrated Energy System Considering Supply-demand UncertaintiesPLOS ONE

Dear Dr. Sun,

Thank you for submitting your manuscript to PLOS ONE. After careful consideration, we feel that it has merit but does not fully meet PLOS ONE’s publication criteria as it currently stands. Therefore, we invite you to submit a revised version of the manuscript that addresses the points raised during the review process.

We look forward to receiving your revised manuscript.

Kind regards,

Zhengmao Li

Academic Editor

PLOS ONE

Journal requirements: When submitting your revision, we need you to address these additional requirements. 1. Please ensure that your manuscript meets PLOS ONE's style requirements, including those for file naming. The PLOS ONE style templates can be found at https://journals.plos.org/plosone/s/file?id=wjVg/PLOSOne_formatting_sample_main_body.pdf and https://journals.plos.org/plosone/s/file?id=ba62/PLOSOne_formatting_sample_title_authors_affiliations.pdf. 2. Please note that PLOS ONE has specific guidelines on code sharing for submissions in which author-generated code underpins the findings in the manuscript. In these cases, we expect all author-generated code to be made available without restrictions upon publication of the work. Please review our guidelines at https://journals.plos.org/plosone/s/materials-and-software-sharing#loc-sharing-code and ensure that your code is shared in a way that follows best practice and facilitates reproducibility and reuse. 3. We note that you have indicated that there are restrictions to data sharing for this study. PLOS only allows data to be available upon request if there are legal or ethical restrictions on sharing data publicly. For more information on unacceptable data access restrictions, please see http://journals.plos.org/plosone/s/data-availability#loc-unacceptable-data-access-restrictions.  Before we proceed with your manuscript, please address the following prompts: a) If there are ethical or legal restrictions on sharing a de-identified data set, please explain them in detail (e.g., data contain potentially identifying or sensitive patient information, data are owned by a third-party organization, etc.) and who has imposed them (e.g., a Research Ethics Committee or Institutional Review Board, etc.). Please also provide contact information for a data access committee, ethics committee, or other institutional body to which data requests may be sent. b) If there are no restrictions, please upload the minimal anonymized data set necessary to replicate your study findings to a stable, public repository and provide us with the relevant URLs, DOIs, or accession numbers. For a list of recommended repositories, please seehttps://journals.plos.org/plosone/s/recommended-repositories. You also have the option of uploading the data as Supporting Information files, but we would recommend depositing data directly to a data repository if possible. We will update your Data Availability statement on your behalf to reflect the information you provide. 4. We note that your Data Availability Statement is currently as follows: [All relevant data are within the manuscript and its Supporting Information files.] Please confirm at this time whether or not your submission contains all raw data required to replicate the results of your study. Authors must share the “minimal data set” for their submission. PLOS defines the minimal data set to consist of the data required to replicate all study findings reported in the article, as well as related metadata and methods (https://journals.plos.org/plosone/s/data-availability#loc-minimal-data-set-definition). For example, authors should submit the following data: - The values behind the means, standard deviations and other measures reported;- The values used to build graphs;- The points extracted from images for analysis. Authors do not need to submit their entire data set if only a portion of the data was used in the reported study. If your submission does not contain these data, please either upload them as Supporting Information files or deposit them to a stable, public repository and provide us with the relevant URLs, DOIs, or accession numbers. For a list of recommended repositories, please see https://journals.plos.org/plosone/s/recommended-repositories. If there are ethical or legal restrictions on sharing a de-identified data set, please explain them in detail (e.g., data contain potentially sensitive information, data are owned by a third-party organization, etc.) and who has imposed them (e.g., an ethics committee). Please also provide contact information for a data access committee, ethics committee, or other institutional body to which data requests may be sent. If data are owned by a third party, please indicate how others may request data access.

Additional Editor Comments:

please revise accordingly

Reviewers' comments:

Reviewer's Responses to Questions

**Comments to the Author**

1. Is the manuscript technically sound, and do the data support the conclusions?

Reviewer #1: Yes

Reviewer #2: Yes

Reviewer #3: Yes

2. Has the statistical analysis been performed appropriately and rigorously? 

Reviewer #1: Yes

Reviewer #2: Yes

Reviewer #3: Yes

3. Have the authors made all data underlying the findings in their manuscript fully available?

Reviewer #1: No

Reviewer #2: No

Reviewer #3: No

4. Is the manuscript presented in an intelligible fashion and written in standard English?

Reviewer #1: Yes

Reviewer #2: Yes

Reviewer #3: Yes

5. Review Comments to the Author

Reviewer #1: This paper developed an optimisation method for district multi-energy systems considering uncertainties. My suggestions are as follows:

1. Research gaps, challenges, and contributions are suggested to be refined in bullet point style. Now the contribution is not clear.

2. Why do authors claim the previous uncertainty modelling in integrated energy systems is not adaptable?

3. Box-type uncertainty modelling has been extensively used before. Please elaborate on how your modelling method is different from previous ones.

4. Are there any decision-making constraints in the planning stage?

5. Please give the reference for the data source (e.g., How the values in Table 1 are determined).

6. Please draw some figures to show the impacts of uncertainties on the investment, rather than only LPSP and other reliability indices.

7. Please include some qualitative results in the conclusion section to be more informative.

Reviewer #2: This paper proposes an optimal sizing method for electricity-heat-hydrogen integrated energy systems. The supply-demand uncertainties are considered in the planning. Although this paper is well-organized, its novelty may be minor. Two comments are listed below.

1) This approach seems like a reproduction of the existing methods, either the optimization model or the solving algorithms. The modification is very marginal.

2) Why Fig. 13 in case studies is blank?

Reviewer #3: This manuscript addresses the problem of optimal capacity allocation for an integrated electricity-heat-hydrogen regional energy system, using a two-stage robust optimization method to construct a capacity allocation model by maximizing and minimizing the worst-case system performance. My comments are as follows:

1. The introduction does not provide a detailed overview of the strengths and weaknesses of the existing literature on specific uncertainty handling methods. Please add a comparative analysis of the applicable scenarios, advantages and disadvantages between robust optimization, interval optimization and stochastic optimization.

2. The literature review is a mixture of technical methods and does not highlight the core research questions and innovations.

3. The range of the “boxed” uncertainty set depends on simple upward and downward biases and does not consider time-series correlations.

4. The manuscript does not discuss the generalization of the model to other energy structures (e.g., a higher proportion of solar energy) or regional conditions (e.g., wet and cold environments in the South).

5. The column constraint generation algorithm was not compared with other possible algorithms, such as mixed-integer linear programming.

6. The results section focuses on total cost and reliability, but there is less analysis of the specific impact of each piece of equipment. Please add an analysis of the independent contribution of changes in capacity configurations of different equipment to economy and reliability.

7. The feasibility of practical implementation of the model was not discussed in the framework of regional policies and the current state of the energy market.

8. The conclusion section only summarizes the research results and does not explore possible directions for future research.

6. PLOS authors have the option to publish the peer review history of their article (what does this mean? ). If published, this will include your full peer review and any attached files.

**Do you want your identity to be public for this peer review?** For information about this choice, including consent withdrawal, please see our Privacy Policy .

Reviewer #1: No

Reviewer #2: No

Reviewer #3: No

---

## [Author Response · Author response to Decision Letter 1]

7 Feb 2025

Thank you for your comments, the detailed revision of this article has been given in the 'Response to Reviewers'.

---

## [Decision Letter · Decision Letter 1]

16 Feb 2025

Capacity Configuration Optimization of Electricity Heat Hydrogen Regional Integrated Energy System Considering Supply-demand Uncertainties

PONE-D-24-55323R1

Dear Dr. Sun,

We’re pleased to inform you that your manuscript has been judged scientifically suitable for publication and will be formally accepted for publication once it meets all outstanding technical requirements.

Kind regards,

Zhengmao Li

Academic Editor

PLOS ONE

Additional Editor Comments (optional):

Reviewers' comments:

Reviewer's Responses to Questions

**Comments to the Author**

1. If the authors have adequately addressed your comments raised in a previous round of review and you feel that this manuscript is now acceptable for publication, you may indicate that here to bypass the “Comments to the Author” section, enter your conflict of interest statement in the “Confidential to Editor” section, and submit your "Accept" recommendation.

Reviewer #2: All comments have been addressed

Reviewer #3: All comments have been addressed

2. Is the manuscript technically sound, and do the data support the conclusions?

Reviewer #2: Yes

Reviewer #3: Yes

3. Has the statistical analysis been performed appropriately and rigorously? 

Reviewer #2: Yes

Reviewer #3: Yes

4. Have the authors made all data underlying the findings in their manuscript fully available?

Reviewer #2: No

Reviewer #3: Yes

5. Is the manuscript presented in an intelligible fashion and written in standard English?

Reviewer #2: Yes

Reviewer #3: Yes

6. Review Comments to the Author

Reviewer #2: Authors have addressed all comments. This manuscript is readily to be accepted. Thanks for the efforts of authors.

Reviewer #3: (No Response)

7. PLOS authors have the option to publish the peer review history of their article (what does this mean? ). If published, this will include your full peer review and any attached files.

**Do you want your identity to be public for this peer review?** For information about this choice, including consent withdrawal, please see our Privacy Policy .

Reviewer #2: No

Reviewer #3: No

---

## [Editor Report · Acceptance letter]

PONE-D-24-55323R1

PLOS ONE

Dear Dr. Sun,

I'm pleased to inform you that your manuscript has been deemed suitable for publication in PLOS ONE. Congratulations! Your manuscript is now being handed over to our production team.

Kind regards,

on behalf of

Dr Zhengmao Li

Academic Editor

PLOS ONE